# Dinickel-catalyzed enantioselective reductive addition of imines with vinyl halides

Peng Zhou[1,5], Peigen Wang[1,5], Hongdan Zhu[2,5], Jian Zhang[1], Qian Peng [2,3] ✉ & Zhonglin Tao [1,4] ✉

A dinickel complex-enabled, enantioselective reductive alkenylation reaction of N-Ts imines with vinyl chlorides is reported. This method is effective with a wide array of (hetero)aryl and aliphatic imines bearing diverse functional groups. Under mildly reductive conditions, synthetically valuable allylic amines are obtained in good yields with excellent enantioselectivities. Furthermore, this catalytic system can be extended to aldehydes, offering an efficient route to access chiral allylic alcohols from readily available starting materials. The potential applications of this method are demonstrated by its use in the late-stage functionalization of biologically active molecular derivatives and the manipulation of reaction products. Initial mechanistic investigations and DFT calculation lend support to a catalytic addition pathway.

α-Chiral amines are ubiquitous in natural products, pharmaceuticals, and other biologically active molecules, making their enantioselective synthesis a critical and enduring pursuit in synthetic chemistry (Fig. 1a)[1–6]. In fact, nearly 40% of the top 200 best-selling small molecule drugs in 2023 contain α-stereogenic amines[7]. Among the diverse methodologies developed for α-chiral amine synthesis, the direct enantioselective addition of imines with carbon nucleophiles stands out as a particularly direct and efficient approach (Fig. 1b). A broad spectrum of carbon nucleophiles, such as organo-magnesium[8–10], -zinc[11–13], -aluminum[14] and -boron[15–19], has been successfully employed in these addition reactions, facilitated by various chiral catalysts. However, the inherent challenges associated with the preparation and handling of these air- and moisture-sensitive organometallic reagents underscore the need for alternative strategies. One such strategy involves the use of readily available, bench-stable carbon electrophiles in direct addition reactions with imines under reductive conditions[20,21]. This approach not only mitigates the practical limitations of organometallic reagents but also offers a more step-economic pathway, given that many organometallic reagents are synthesized from carbon electrophiles.

Asymmetric reductive carbonyl addition reactions with various carbon electrophiles have been extensively studied and notable advancements have been achieved through various transition-metal catalysis, such as chromium (known as the Nozaki–Hiyama–Kishi (NHK) reaction)[22–26], nickel[27–32], and cobalt[33–38]. In contrast, the exploration of analogous reductive addition reactions of imines remains limited (Fig. 1c). For instance, in 2023, the Zhou group innovatively reported a nickel-catalyzed enantioselective reductive addition of aldimines with a wide range of aryl halides and sulfonates, utilizing the chelating properties of an azaaryl protecting group, which is not readily deprotected[39]. Concurrently, the Shi group disclosed a chiral cobalt-bisphosphine catalyst enabled the asymmetric reductive (hetero)arylation of cyclic N-sulfonyl imines with aryl halides, though this method was restricted to cyclic imine substrates[40]. The Chen group further contributed by reporting a cobalt-catalyzed enantioselective reductive vinyl addition of α-imino esters with alkenyl halides, leading to the formation of enantioenriched α-vinylic amino acids featuring quaternary carbon centers[41–44]. Despite these efforts, transition-metal-catalyzed aza-NHK-type reactions, particularly those involving simple acyclic N-sulfonyl imines, remain underexplored due to several challenges: 1) Limited bonding-site of imine which can't form chelation interaction with catalyst causes difficulty for stereoinduction; 2) Simple imines are unable to form favorable 6-membered cyclic

[1]State Key Laboratory of Chemo and Biosensing, College of Chemistry and Chemical Engineering, Hunan University, Changsha, China. [2]State Key Laboratory of Elemento-Organic Chemistry, Frontiers Science Center for New Organic Matter, College of Chemistry, Nankai University, Tianjin, China. [3]Tianjin Key Laboratory of Biosensing and Molecular Recognition, Tianjin, China. [4]Greater Bay Area Institute for Innovation, Hunan University, Guangzhou, China. [5]These authors contributed equally: Peng Zhou, Peigen Wang, Hongdan Zhu. ✉e-mail: qpeng@nankai.edu.cn; taozl@hnu.edu.cn

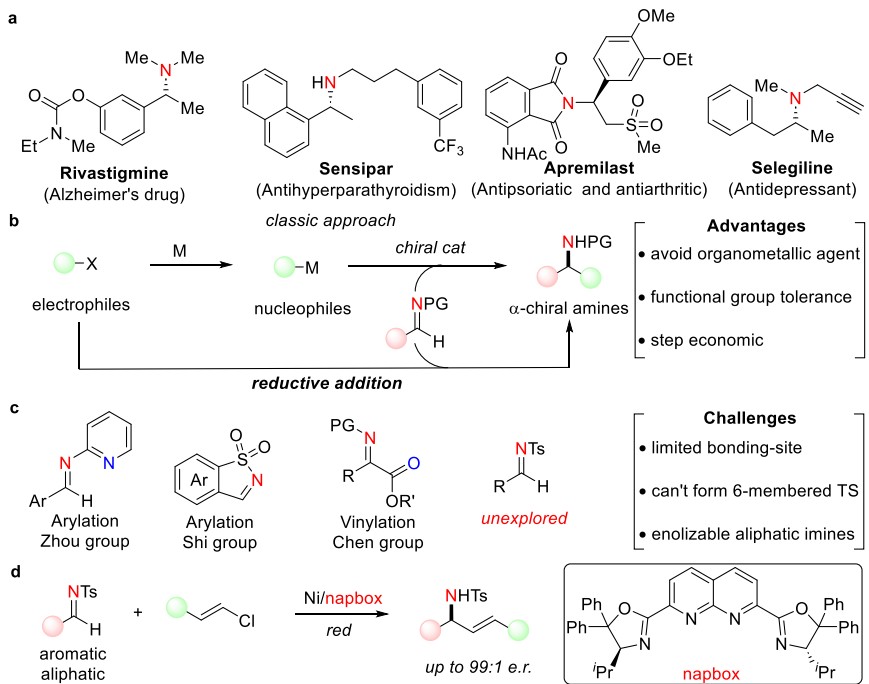

**Fig. 1 | Introduction for asymmetric reductive addition of imines. a** Biologically active α-chiral amines. **b** Comparison between classic organometallic addition and reductive addition to imines. **c** Specific imines have been successfully applied in reductive addition reactions and the challenges of simple *N*-protected imines. **d** Dinickel-catalyzed reductive addition of *N*-Ts imines with vinyl chlorides (this work).

transition states; 3) Aliphatic imines easily undergo isomerization to unreactive enamines.

Our previous work demonstrated that a dinickel complex[45–47] could catalyze the enantioselective α-alkylation of acyclic ketones using unactivated alkyl iodides[48]. Mechanistic investigations revealed that the low-valent organonickel species generated in this system exhibited significant nucleophilicity, consistent with other findings[27–32,49–51]. Motivated by these observations, we hypothesized that this dinickel complex could also facilitate the reductive addition of imines with vinyl halides (Fig. 1d). Success in this approach would enable the efficient synthesis of enantiomerically enriched allylic amines[52–56], valuable intermediates in organic synthesis.

## Results

### Optimization reaction conditions

We initiated our investigation by examining the reductive alkenyl addition of *N*-tosyl imine **1a** with styrenyl chloride **2a**. The reaction was conducted in the presence of NiBr$_2$·glyme as the catalyst, zinc dust as the reductant, and sodium iodide (NaI) as an additive, using a selection of bimetallic naphthyridine-bis(oxazoline) (napbox) ligands[46,57] in tetrahydrofuran (THF) (Table 1, entries 1–3). Initial screening identified several promising ligands (**L1–L3**) that facilitated the reaction with moderate yields and high enantioselectivities. Among these, napbox **L1** provided the highest enantioselectivity (entry 1). Extending the reaction time to 48 h increased the yield to 79% (entry 4). However, employing Mn instead of Zn as the reductant resulted in a slight reduction in both yield and enantioselectivity (entry 5). Notably, increasing the reaction concentration further improved the yield to 91% without affecting enantioselectivity (entry 6). Control experiments showed the significant role of NaI in the reaction to receive a high yield (entry 7). Despite establishing optimal conditions, we also evaluated several monomeric ligands commonly used in combination with Ni catalysts (entries 8–14). The Box ligand **L4** and BiOx ligand **L5** were ineffective in this transformation (entries 8–9). Additionally, Pyox **L7**, Quinox **L8**, Napox **L9**, and Pybox **L10**, which possessed the same oxazoline ring substitutions as **L1**, produced the addition products

with markedly lower efficiencies (entries 11–14). These results underscored the superior performance of bimetallic ligands in this transformation.

### Substrate scope

Under the optimized reaction conditions, we explored the substrate scope of this transformation, as summarized in Fig. 2. Initially, imines bearing various sulfonyl protecting groups underwent reductive vinylic addition with high yields, though the choice of protecting group notably influenced enantioselectivity (**4–8**). Subsequently, a range of imines, including both aromatic and aliphatic variants, were subjected to the reaction (**9–33**). Aromatic imines with substituted phenyl rings generally provided the corresponding allylic amines in good yields (**9–19**), though *ortho*-substitution led to reduced yield and enantioselectivity (**13**). Imine bearing a strongly electron-deficient substituent showed lower conversion, albeit with preserved enantioselectivity (**19**). Additionally, naphthyl and heteroaryl imines proved to be compatible with the reaction conditions, affording chiral amines in moderate to high yields and excellent enantioselectivities (**20–25**). The reaction also demonstrated broad tolerance toward aliphatic imines, which were typically challenging due to undesired deprotonation when using organometallic reagents, yielding allylic amines in high yields with remarkable enantioselectivities (**26–33**). Finally, the scope of vinyl chlorides was examined (**34–43**). Various *trans*-styrenyl chlorides with different substitutions on the phenyl ring reacted efficiently, regardless of the substitution patterns (*ortho*, *meta*, or *para*) and electronic properties (electron-withdrawing or -donating). Notably, aryl bromide, which offered further functionalization opportunities, remained intact (**38**). However, *cis*-styrenyl chloride and alkyl-substituted vinyl chloride were unsuitable substrates for this transformation (see Supplementary Information (SI) for details).

Furthermore, the dinickel catalyst also proved effective in the reductive vinylic addition of aldehydes, which were less electrophilic compared to *N*-Ts imines. Recently, both the Meng[36] and Shi[29] groups independently reported enantioselective reductive alkenylation of aldehydes using chiral Co and Ni catalysts, respectively. However, in

**Table 1 | Optimization of reaction conditions[a]**

| Entry | L | X | Yield (%) | e.r. |
|---|---|---|---|---|
| 1[b,c] | L1 | 5 | 61 | 98:2 |
| 2[b,c] | L2 | 5 | 61 | 93:7 |
| 3[b,c] | L3 | 5 | 53 | 93:7 |
| 4[c] | L1 | 5 | 79 | 98:2 |
| 5[c,d] | L1 | 5 | 42 | 95:5 |
| 6 | L1 | 5 | 91 | 98:2 |
| 7[e] | L1 | 5 | 54 | 98:2 |
| 8 | L4 | 10 | trace | – |
| 9 | L5 | 10 | trace | – |
| 10 | L6 | 10 | trace | – |
| 11 | L7 | 10 | 28 | 50:50 |
| 12 | L8 | 10 | 13 | 53:47 |
| 13 | L9 | 10 | 28 | 62:38 |
| 14 | L10 | 10 | 55 | 62:38 |

[a]Unless noted, imine **1a** (0.2 mmol), styrenyl chloride **2a** (1.5 equiv), NiBr$_2$·glyme (10 mol%), **L** (x mol%), NaI (2.0 equiv) and Zn (2.0 equiv) were stirred in THF (0.2 M) at 25 °C for 48 h, isolated yield.
[b]Reaction with **2a** (1.2 equiv) for 24 h.
[c]0.1 mmol scale in THF (0.1 M).
[d]Mn was used instead of Zn.
[e]Without NaI.

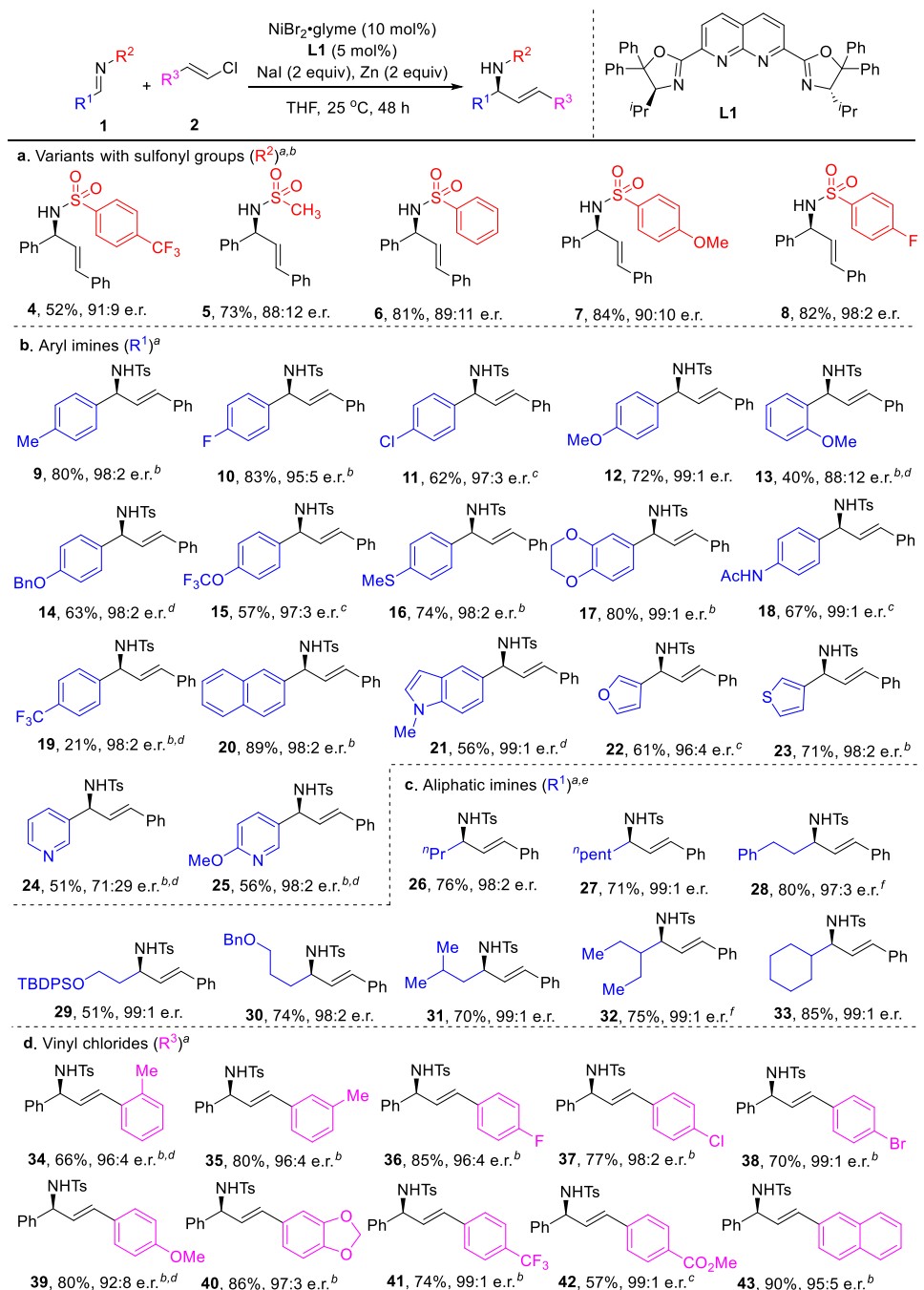

**Fig. 2 | Substrate scope of reductive alkenylation of imines. a** Scope of different sulfonyl protecting groups. **b** Scope of aryl imines. **c** Scope of aliphatic imines. **d** Scope of vinyl chlorides. [a]Reaction condition as shown in Table 1. [b]ZnBr$_2$ (20 mol%) was added. [c]Imine **1** (1.5 equiv), vinyl chloride **2** (0.2 mmol) was reacted in THF (0.2 M) at 40 °C for 48 h. [d]40 °C, 48 h. [e]ZnBr$_2$ (1.0 equiv) was added, **L3** as ligand, in THF (0.1 M) at 25 °C for 24 h. [f]40 °C for 24 h. TBDPS *t*-butyl-diphenylsilyl.

the Ni-catalyzed system, only moderate enantioselectivity (71:29 e.r.) was achieved with *trans*-styrenyl halides when using a monomeric ligand. In contrast, employing the bimetallic ligand (see SI for the optimization) resulted in significantly enhanced enantioselectivity with this type of substrates (Fig. 3). A diverse range of substituted benzaldehydes (**44–58**), irrespective of the electronic nature of the substitution (electron-donating or withdrawing) and substitution patterns (*ortho*, *meta*, *para*), naphthyl (**59**) and heteroaryl aldehydes (**60–63**) underwent reductive transformation efficiently, yielding allylic alcohols in moderate to high yields with excellent enantioselectivities. Furthermore, the reaction exhibited broad functional group tolerance, including aryl bromide (**48**), aryl iodide (**49**), ether

(**50–52**), thioether (**54**), boronic ester (**55**), ester (**57**), cyano (**58**), and carbamate (**62**) functionalities. However, when aliphatic aldehydes were used as substrates, the reaction afforded low yields, though enantioselectivity was retained (see SI for details). This observation aligned with previous reports on Ni-catalyzed reductive addition reactions of aldehydes[27–29].

## Synthetic applications

To demonstrate the practical utility of this methodology, several key transformations were carried out (Fig. 4). First, enantioselective reductive vinylic addition reactions were performed on aldehydes derived from various biologically active compounds, enabling the

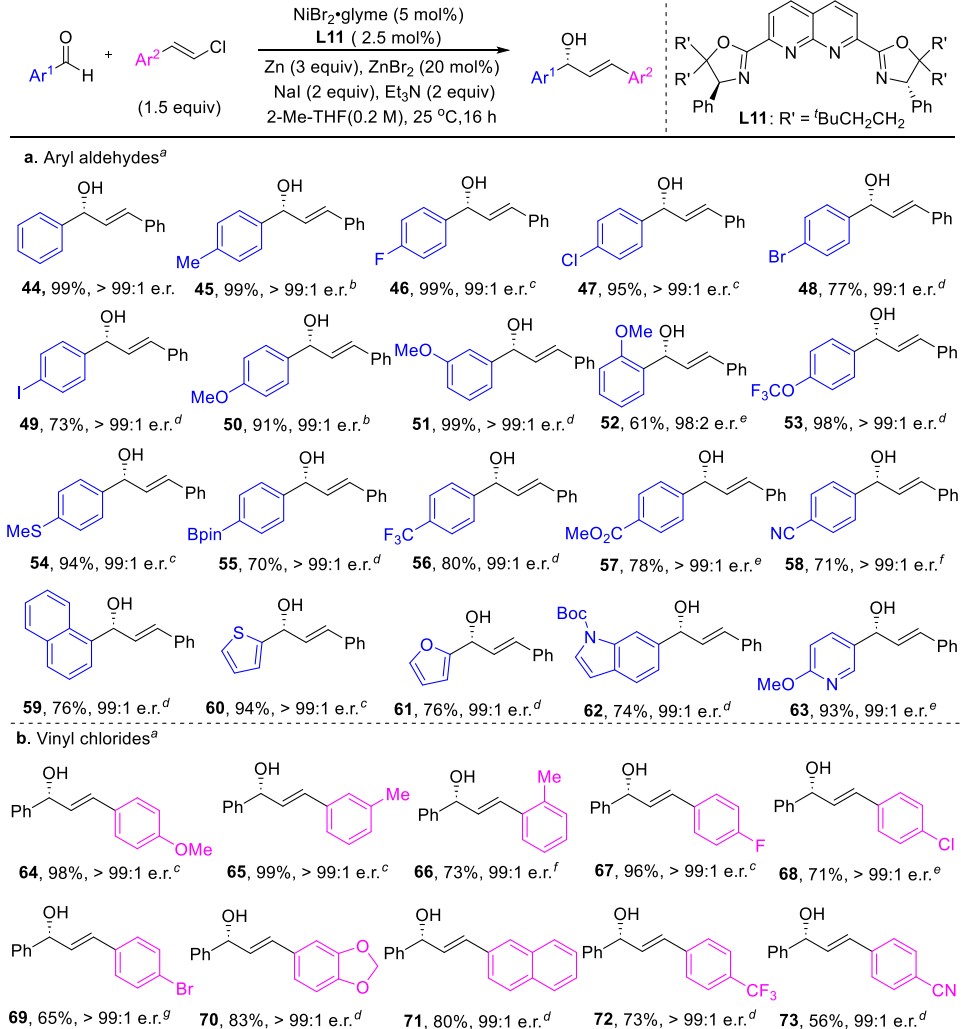

**Fig. 3 | Substrate scope of reductive alkenylation of aldehydes. a** Scope of aldehydes. **b** Scope of vinyl chlorides. [a]Under N₂, aldehydes (0.2 mmol, 1.0 equiv), vinyl chloride **2** (1.5 equiv), NiBr₂•glyme (5 mol%), **L11** (2.5 mol%), NaI (2.0 equiv), ZnBr₂ (20 mol%), NEt₃ (2.0 equiv) and Zn (3.0 equiv) were stirred in 2-Me-THF (0.2 M) at 25 °C for 16 h. [b]24 h. [c]NiBr₂•glyme (10 mol%), **L11** (5 mol%), 16 h. [d]NiBr₂•glyme (10 mol%), **L11** (5 mol%), 24 h. [e]NiBr₂•glyme (10 mol%), **L11** (5 mol%), 48 h. [f]NiBr₂•glyme (10 mol%), **L11** (5 mol%), 40 °C, 24 h. [g]NiBr₂•glyme (10 mol%), **L11** (5 mol%), 0 °C, 48 h. Bpin pinacolato-boron. Boc *t*-butyloxy carbonyl.

late-stage installation of allylic alcohols with excellent diastereoselectivities under mild conditions (Fig. 4a). Additionally, the chiral allylic amines generated from the reductive vinylic addition of imines proved versatile for further functionalizations (Fig. 4b). The tosyl protecting group was readily removed, affording primary amine **78** with high enantiospecificity. Moreover, *N*-tosyl amine **3** was efficiently converted to *N*-Boc amine **79**, highlighting the ease of nitrogen elaboration in allylic amines. The olefin functionality also demonstrated significant reactivity, undergoing hydrogenation to yield **80** and epoxidation to produce **81**, both with retention of enantioselectivities. Notably, reaction of product **9** with bis(collidine)bromonium(I) hexafluorophosphate (BBH) furnished azetidine **82**, containing three contiguous stereocenters. These examples underscored the synthetic versatility and broad applicability of the products obtained via this methodology.

## Mechanistic study

To gain insights into the reaction mechanism, several experiments were conducted (Fig. 5). First, to determine whether the active catalytic species was bimetallic or monomeric, we synthesized a dinickel complex stabilized by an acetate ion following the method reported by the Uyeda group[46]. This dinickel complex efficiently catalyzed the reductive

alkenylation of imines, producing the desired product with enantioselectivity comparable to that observed in the original reaction (Fig. 5a). This result strongly suggested that the dinickel species functioned as the active catalyst. Regarding the C−C bond formation, two potential mechanisms were considered: (a) a Ni-catalyzed cross-coupling pathway, where the imine underwent single-electron reduction to generate an α-amino radical[58–60] that subsequently recombined with Ni, followed by reductive elimination, and (b) a 1,2-addition process. To rule out the cross-coupling mechanism, we introduced a radical probe as an additive, but no radical adduct was detected (Fig. 5b). Instead, the imine alkenylation product was formed with high efficiency. Additionally, when an olefin-tethered imine **2b** was subjected to the reaction conditions, no cyclization product was observed, and the direct addition product **83** was isolated in moderate yield with high enantioselectivity (Fig. 5c). Moreover, when cyclopropyl imine **1c** was employed, the desired addition product **84** was isolated, without observation of ring-opening cross-coupling product. These findings were inconsistent with a radical cross-coupling mechanism. In order to support a 1,2-addition pathway, a competition reaction between imine and aldehyde was conducted under the reductive alkenylation condition (Fig. 5d). The addition to more electrophilic imine predominated the reaction, consistent with 1,2-addition mechanism.

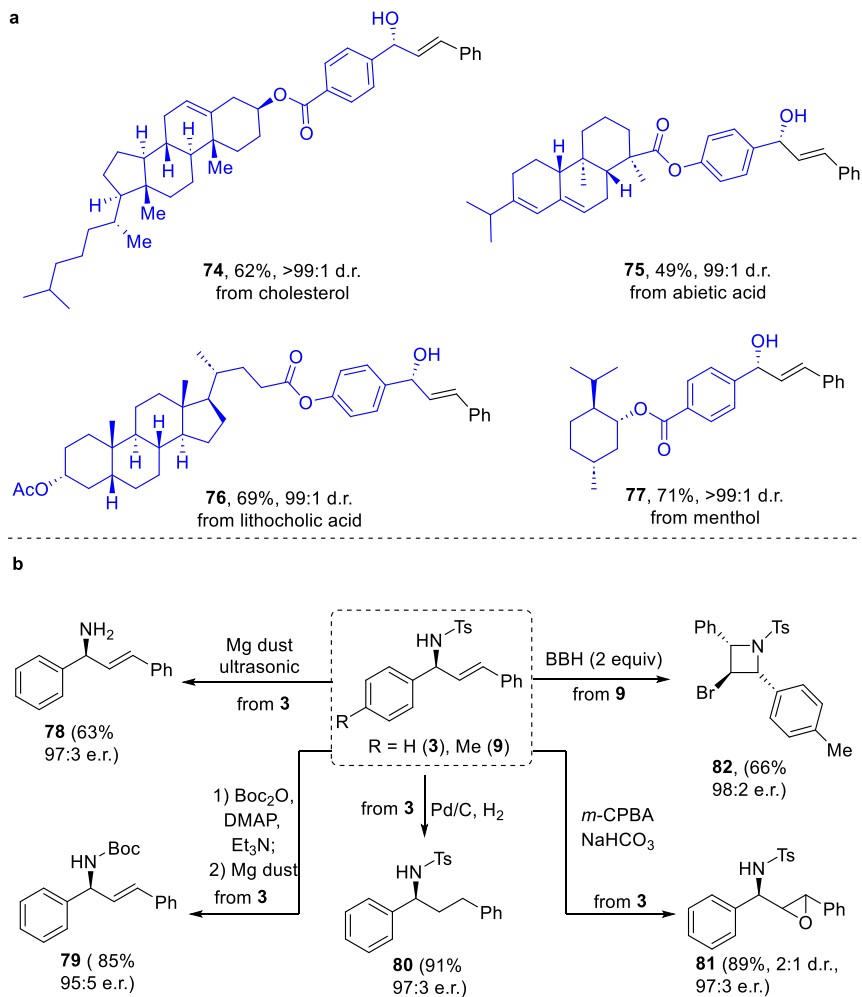

**Fig. 4 | Synthetic applications. a** Late-stage functionalization of complex aldehydes. **b** Product manipulation on allylic amines.

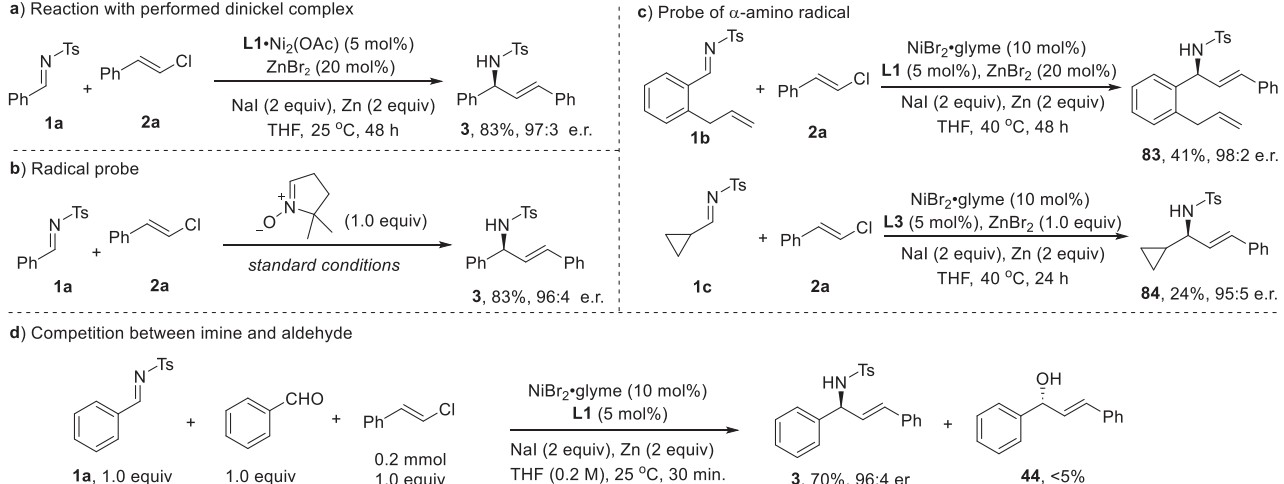

**Fig. 5 | Mechanistic study. a** Reaction catalyzed by performed dinickel complex. **b** Radical probe experiment. **c** Ring-closing and opening experiments. **d** Competing experiment between aldehyde and imine.

## DFT calculation

To further unveil the enantioselective mechanism of the dinickel-catalyzed reductive addition of imines with vinyl halides, unrestricted density functional theory calculations were performed at SMD-M06L-D3/def2-TZVP//M06L/def2-TZVP/def2-SVP level of theory (see computational details and benchmark test in SI and Supplementary Data 1). The different spin states of calculated models had been evaluated for the energy profile, indicating the reaction mechanism should majorly occur in the favorable triplet states, although selected intermediates might transfer to quintet state as shown in Supplementary

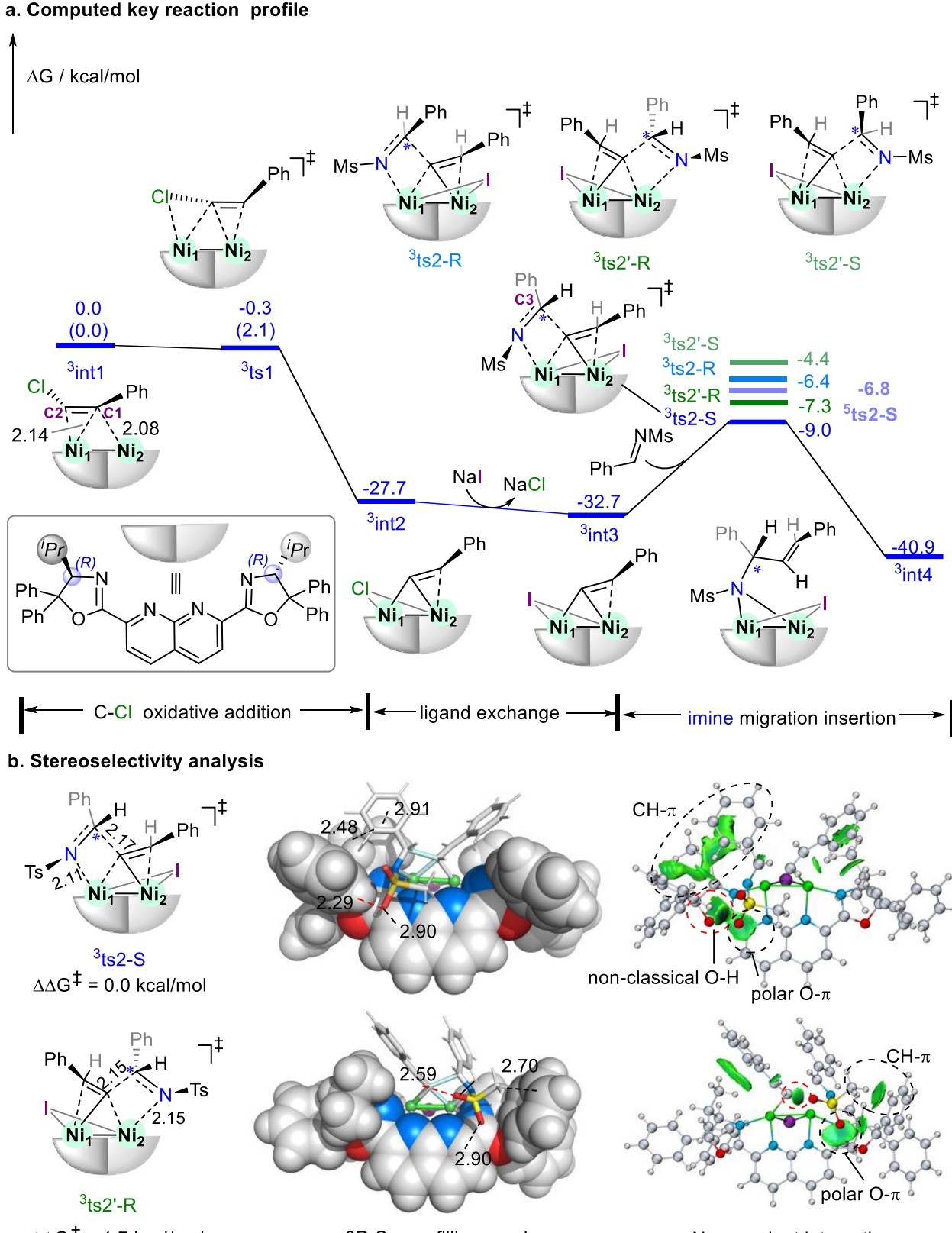

**Fig. 6 | Calculated enantioselective mechanism of the dinickel-catalyzed reductive addition of imines with vinyl halides. a** Computed key reaction profile. **b** Stereoselectivity analysis. Bond length (Å) in black. IGMH analysis of noncovalent interaction with δg$^{inter}$ = 0.004. The values in parentheses represent the electron energy.

Fig. S8. The triplet state of dinuclear (napbox)Ni$_2$ ($^3$[Ni]$_2$) as its ground state can facilitate the activation of styrenyl chloride **2a** to form a stable π-complex with C1 coordinated to both Ni1 and Ni2, supporting by an exothermic energy of −43.7 kcal/mol (Supplementary Fig. S7). Thus, this predominant dinickel π-complex $^3$**int1** achieves the favorable energy profile as shown the key pathway in Fig. 6a. Furthermore, the following oxidative addition of the C−Cl bond give a barrierless step via $^3$**ts1**, indicating the high reactivity of the triplet species. Both Ni co-participate in the oxidative addition generated the bridging species $^3$**int2** via the vinyl and chloride group. Under the excess NaI, the halide anion exchange occurs from Cl$^-$ to I$^-$ driven by favorable thermodynamic energy (−5.0 kcal/mol in triplet states). The formed iodide intermediate $^3$**int3** undergoes the direct imine migration insertion of *N*-Ms imine by a majorly reactive Ni and a synergistic Ni. This enantioselective step is determined by the addition from the *Re* or *Si* face of the imine. Calculated results indicate that the *Si* face attacking majorly on the Ni1 reaction site achieve the most stable transition state $^3$**ts2-*S*** (ΔG$^‡$ = 23.7 kcal/mol, 2.2 kcal/mol lower than that of $^5$**ts2-*S***), leading to *S*-configured intermediate $^3$**int4** in agreement with the experimental observations *S*-allylic amines. The other *Re* face attacking and the alternative Ni2 reaction site have also been evaluated as shown in $^3$**ts2-*R***, $^3$**ts2'-*R*** and $^3$**ts2'-*S***, respectively (the energy profile dominated by the Ni2 reaction site as shown in Supplementary Fig. S9). Considering the Boltzmann distribution, the predicted ee value is 87% (exp. 76%). A closer inspection of the most corresponding $^3$**ts2-*S*** and $^3$**ts2'-*R*** indicates that Ni1 as the reactive site can strongly bind with the imine substrate and facilitate the addition comparing with Ni2 site (2.11 Å in $^3$**ts2-*S*** vs. 2.15 Å in $^3$**ts2'-*R***). The relatively early transition state in $^3$**ts2-*S*** further support the high reactivity (C2-C3: 2.17 Å). For the chiral control, there are apparently CH-π, polar O-π and non-classical O-H weak interactions between the imine and the chiral ligand in $^3$**ts2-*S***, supporting by the spatial distances and IGMH analysis. The illustrated 3D space-filling mode also shows the matched chiral cavity of catalyst with the reactive substrates in $^3$**ts2-*S***. Benefited from the binuclear Ni complex, their synergistic interaction enhances the reactivity and enantio-selectivity. The *R, R*-napbox ligand achieved the pronounced weak interaction with imine substrate to regulate the *S*-enantioselectivity.

Our plausible mechanism was depicted in Fig. 7. Active π-complex **int1**, formed via the coordination of styrenyl chloride **2a** with dinuclear Ni, undergoes C−Cl bond oxidative addition to produce a vinyl-bridged intermediate **int2**. After the ligand exchange by NaI to the iodide intermediate **int3**, the nucleophilic addition of the *Si*-face imine substrate controlled by *R, R*-napbox ligand occurs to generate the *S*-intermediate **int4**. Finally, releasing addition product via ligand exchange by ZnI$_2$ can form **int5**, which undergo the reduction by Zn and the coordination of another **2a** to regenerate **int1** for the next catalytic cycle.

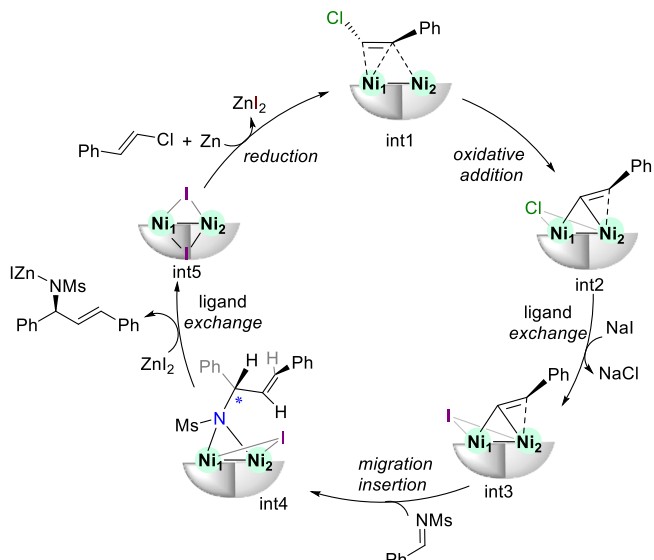

**Fig. 7 | Proposed mechanism.** Proposed catalytic cycle for the dinickel-catalyzed reductive alkenylation of imines with vinyl halides.

## Discussion

In summary, we have developed a dinickel-catalyzed reductive alkenylation reaction of imines and aldehydes using vinyl chloride as the alkenylation agent. This approach enables the efficient synthesis of allylic amines and allylic alcohols in good yields with excellent enantioselectivities. The mild reaction conditions exhibit broad functional group tolerance, accommodating substrates that are often incompatible with traditional organometallic reagents. The unique efficacy of the dinickel catalyst, particularly in suppressing side reactions and enhancing chiral induction, underscores the utility of bimetallic systems in asymmetric catalysis.

## Methods
### Representative procedure for the synthesis of compound 3
The NiBr$_2$•glyme (6.1 mg, 0.02 mmol, 10 mol%), **L1** (6.6 mg, 0.01 mmol, 5 mol%), NaI (60.0 mg, 0.4 mmol, 2.0 equiv) and Zn powder (26.0 mg,

0.4 mmol, 2.0 equiv) were introduced into a flame-dried Schlenk tube in a N$_2$-filled glove box. After removal from the glove box, Schlenk tube was connected to Schlenk line under N$_2$. Dry THF (1 mL) was injected into the Schlenk tube, the mixture was stirred at r.t. for 30 min. The vinyl chloride (0.3 mmol, 1.5 equiv) and imine (0.2 mmol, 1.0 equiv) were sequentially added under nitrogen. The resulting mixture was stirred under particular temperature for 48 h. The reaction mixture was cooled to room temperature, quenched with saturated NH$_4$Cl (aq.) solution and extracted with EtOAc (3x). The organic phase was washed with brine, dried over anhydrous Na$_2$SO$_4$, filtered, and concentrated under reduced pressure. The crude material was purified using column chromatography to give the pure product.

## Data availability
The data reported in this paper are available within the article and its Supplementary Information files (including experimental details, NMR data, computational details, NMR and HPLC spectrums). All data are also available from the corresponding author upon request.

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

## Acknowledgements

The authors are grateful for the financial support from the National Key R&D Program of China (2021YFA1500100, Q.P.), the Natural Science Foundation of China (22471067, Z.T., 92156017, Q.P. and 22403053, H.Z.), the Natural Science Foundation of Hunan Province (2022JJ20006, Z.T.), Natural Science Foundation of Tianjin (24JCZDJC00750, Q.P.), "Frontiers Science Center for New Organic Matter", Nankai University (63181206, Q.P.), Haihe Laboratory of Sustainable Chemical Transformation of Tianjin (24HHWCSS00019, Q.P.), the Postdoctoral Fellowship Program of China Postdoctoral Science Foundation (GZC20240750, H.Z.) and the Fundamental Research Funds for the Central Universities. We also thank the Analytical Instrumentation Center of Hunan University for mass spectrometry analysis.

## Author contributions

Z.T. conceived and designed the project. P.Z., P.W. and J.Z. conducted the experiments. H.Z. conducted the DFT calculations. Q.P. and Z.T. wrote the manuscript. All authors contributed to analyse the data and edit the manuscript.

## Competing interests

The authors declare no competing interests.
