## [Transparent Peer Review file · Nature Communications]

Dinickel-Catalyzed Enantioselective Reductive Addition of Imines with Vinyl Halides

Corresponding Author: Professor Zhonglin Tao

Version 0:

Reviewer comments:

Reviewer #1

(Remarks to the Author)

This paper described a Napbox/nickel-catalyzed highly enantioselective addition of vinyl chlorides to aldehydes and aldehyde imines in the presence of zinc, which provided chiral allylic alcohols and amines with good yields as well as good functional group tolerance. However, the asymmetric reductive coupling between vinyl electrophile and imines has been recently reported by Chen and Wu. The asymmetric reductive vinyl addition of aldehydes was also achieved by Meng and Shi. On the other hand, the PHOX/Cobalt-catalyzed asymmetric reductive coupling between alkynes and imines enables synthesis of chiral allylic amines nearly the same with this paper. (J. Am. Chem. Soc. 2023, 145, 20152–20157) So, the novelty of this work doesn't match the standard of Nature Communication. I suggest the paper be not suitable to publish on this magazine.

Other comments:

- 1, the language should be polished, such as " However, substituting Zn with Mn....."
- 2, The authors in references should be fully listed.

Reviewer #2

(Remarks to the Author)

The present manuscript constitutes an excellent piece of interesting and well-conducted research, which is also clearly presented. The authors have developed an enantioselective addition of vinyl halides to imines and aldehydes. Worthy of special mention are the extensive optimization efforts, including a good number of asymmetric ligands. In this regard, it is somewhat disappointing to see the different outcomes obtained for the imine and aldehyde substrates. After identifying the best conditions for imines—with a maximum of 96% ee—the authors re-optimized the reaction conditions for aldehydes and discovered a new ligand, L11, which delivers almost perfect yields and enantioselectivities. Surprisingly, as far as I can tell, ligand L11 was not tested with the imine substrates. While it is understandable that time constraints often influence the scope of optimization studies, it would be valuable to assess the performance of L11 with at least a couple of imine substrates, given its promising potential under the newly optimized conditions. The authors have also experimentally demonstrated the participation of dinickel species in the reaction, which were then used in the theoretical calculations. DFT calculations have been carried out to explain the mechanism and selectivity of the reaction, and these have been conducted with care. My only question in this section concerns the role of NaI. As much as 2 equivalents of NaI seem to be optimal, but its role has not been commented on or assessed. In the calculations, a halide exchange is proposed from Int2 to Int3. Is this based on a previous hypothesis? Are there relevant literature references that should be included? Why has this specific point in the mechanism been chosen for the exchange? Other minor concerns relate to the energy differences between the four diastereomeric transition states, especially between ts_{2-S} and $ts_{2'-R}$, which are only 1.7 kcal/mol apart—a bit short to fully explain the high enantioselectivity. According to the authors' own calculations, the Boltzmann distribution yields a computed 87% ee (14:1), which is not as close energetically to the experimental 96% ee (49:1) as the authors suggest. This level of disagreement is understandable, given the nature of the species and methods used, but the agreement should not be oversold. Finally, intermediate Int-5 in Figure 7 should contain two iodine atoms attached to the Ni–Ni complex. These concerns do not affect the overall high quality of the work and manuscript. I therefore recommend publication of the article after minor revision, based on my previous comments.

Reviewer #3

(Remarks to the Author)

Although asymmetric reductive addition of imines and aldehydes has been in development for many years, this strategy is still limited to substrates with different modes of substitution. Synthetic chemists are pursuing and expecting a more general approach. Faced with this problem, Tao reported an efficient dinickel catalytic system with high substrate compatibility is achieved. We were surprised by the significant e.r. value advantage of this system compared to mononuclear nickel. The mechanism study and DFT study revealed that this selectivity comes from the fact that the binuclear nickel can bind the substrate tightly to the reaction core through additional Ni-substrate interaction, reducing its spatial freedom, and the asymmetric ligand-substrate interaction ultimately determines its selectivity. However, we think that there are still problems in DFT calculation, and we hope author revise it and then publish it on Nat. Commun.

Here are our suggestions:

1. The PES of the exchange of NaCl and NaI is marked as -3.2 kcal/mol. However, due to the absence of the SPE and coordinates of I source and Cl source in SI, we cannot deduce this energy, and there is also a lack of reasonable literature citations to show that this energy decreases.
2. Considering that this reaction can be carried out without sodium iodide, the need for iodide to replace chloride in this step is questionable, and it should be explained whether the free energy change in ts2 is higher in the case of chlorine-coordination.
3. Are there imine-coordinated intermediates before ts2? Its free energy may be lower than -30.9 kcal/mol, which is crucial to understanding the step energy of 1,2-addition with the dinuclear nickel species.
4. The paper cited in SI (ACS Catal. 2017, 7, 7, 4796–4804) considers a quintet path under a similar system. Whether this path is possible in the reaction presented in this paper, we think it is necessary to add DFT calculations or related literature to illustrate this point.

Here are two minor problems:

1. 3in5 is missing in the potential energy surface.
2. Many intermediates and transition states in Figure 6 lack ΔE value and have unreasonable relative energy heights (an example: "-0.3" is higher than "0")

Version 1:

Reviewer comments:

Reviewer #2

(Remarks to the Author)

The authors have made the necessary changes, and the article can be published as it is now.

Reviewer #3

(Remarks to the Author)

Now the paper can be accepted

We think all the reviewers for the valuable comments and suggestions.

Reply to comments by Referee 1

General Comments: This paper described a Napbox/nickel-catalyzed highly enantioselective addition of vinyl chlorides to aldehydes and aldehyde imines in the presence of zinc, which provided chiral allylic alcohols and amines with good yields as well as good functional group tolerance.

Comment 1: However, the asymmetric reductive coupling between vinyl electrophile and imines has been recently reported by Chen and Wu.

- Indeed, Chen and Wu successfully reported the asymmetric reductive coupling between vinyl electrophile and **α -imino esters**, providing an excellent method to access chiral α -alkenyl quaternary amino esters (shown in Figure 1c). However, in our view, their catalytic system is not compatible with more general acyclic *N*-Ts imines. As mentioned in the introduction, this type of imines presents a significant challenge due to lack of chelation interactions with chiral catalysts. To the best of our knowledge, our manuscript presents the **first** enantioselective NHK-type reaction employing **acyclic *N*-Ts imines**. Moreover, this work represents the first application of a **dinickel catalyst** in an enantioselective NHK-type reaction.

Overall, in our opinion, this work exhibits two major advances: the breakthrough in engaging more challenging and general substrate and new application of dinickel catalyst in enantioselective aza-NHK reaction.

Comment 2: The asymmetric reductive vinyl addition of aldehydes was also achieved by Meng and Shi.

- We agree that the asymmetric reductive vinyl addition of aldehydes has been

previously reported by the Meng group using Co catalyst (cited as ref. 36) and the Shi group using Ni catalyst (cited as ref. 29). However, it's interesting to make a narrow comparison within Ni catalyst. Only moderate enantioselectivity is obtained with *trans*-styrenyl halides as reactant in Shi's work, while excellent enantioselectivity is realized with our dinickel system. This difference highlights the unique reactivity of bimetallic catalyst in asymmetric reductive addition of carbonyl compounds—an area that, to date, remains poorly understood.

Comment 3: On the other hand, the PHOX/Cobalt-catalyzed asymmetric reductive coupling between alkynes and imines enables synthesis of chiral allylic amines nearly the same with this paper. (*J. Am. Chem. Soc.* **2023**, *145*, 20152–20157) So, the novelty of this work doesn't match the standard of Nature Communication. I suggest the paper be not suitable to publish on this magazine.

- The work reported by the Uyeda group (*J. Am. Chem. Soc.* **2023**, *145*, 20152–20157, cited as ref. 56) is an efficient tool to synthesize valuable chiral allylic amines, the same products produced by our method. However, we think they are different in the following aspects.

Starting material: Uyeda's work uses terminal alkynes as substrate, while our work starts with vinyl chlorides.

Reaction mechanism: The reaction reported by Uyeda group undergoes a Co/PHOX-catalyzed oxidative cyclometallation of alkynes and imines pathway, while our reaction is a dinickel-catalyzed 1,2-addition of imine by vinyl nickel species.

oxidative cyclometallation

1,2-addition

Scope: Since these are two different reactions, the chiral allylic amines obtained can be complementary to each other. About the imines, aliphatic imines are not reactive in Uyeda's work, while our reductive addition method is compatible with various aliphatic *N*-Ts imines (shown in Figure 1c).

About the alkenyl side, our method is also effective with electron-deficient substituted styrenyl group. However, this type of product is not demonstrated in Uyeda's work.

Overall, these two methods provide two different tools to access the highly valuable chiral allylic amines.

Other comments:

Comment 4: 1, the langue should be polished, such as " However, substituting Zn with Mn....."

- It has been corrected to "However, employing Mn instead of Zn.....".

Comment 5: 2, The authors in references should be fully listed.

- For the format request of reference in Nat. Commun., the omission is required

when more than 5 authors are listed.

Reply to comments by Referee 2

General Comments: The present manuscript constitutes an excellent piece of interesting and well-conducted research, which is also clearly presented. The authors have developed an enantioselective addition of vinyl halides to imines and aldehydes. Worthy of special mention are the extensive optimization efforts, including a good number of asymmetric ligands. In this regard, it is somewhat disappointing to see the different outcomes obtained for the imine and aldehyde substrates. After identifying the best conditions for imines—with a maximum of 96% ee—the authors re-optimized the reaction conditions for aldehydes and discovered a new ligand, **L11**, which delivers almost perfect yields and enantioselectivities.

➤ We appreciate this referee for these positive comments.

Comment 1: Surprisingly, as far as I can tell, ligand **L11** was not tested with the imine substrates. While it is understandable that time constraints often influence the scope of optimization studies, it would be valuable to assess the performance of **L11** with at least a couple of imine substrates, given its promising potential under the newly optimized conditions.

➤ We conducted the reductive vinylic addition reaction with ligand **L11** shown below. Decreased enantioselectivities and yields were observed. In fact, at the early optimization stage, we noticed ligand with alkyl group at 2-position of oxazoline ring provided better performance in the reductive addition reaction with imines, while ligand with aryl group at 2-position of oxazoline ring was effective for the reductive addition to aldehydes. Considering the differences between imine and aldehyde, the switch of ligands could achieve the highly enantioselective reductive vinylic addition to both of them.

General Comments: The authors have also experimentally demonstrated the participation of dinickel species in the reaction, which were then used in the theoretical calculations. DFT calculations have been carried out to explain the mechanism and selectivity of the reaction, and these have been conducted with care.

➤ We appreciate this referee for these positive comments.

Comment 2: My only question in this section concerns the role of NaI. As much as 2 equivalents of NaI seem to be optimal, but its role has not been commented on or assessed. In the calculations, a halide exchange is proposed from **Int2** to **Int3**. Is this based on a previous hypothesis? Are there relevant literature references that should be included? Why has this specific point in the mechanism been chosen for the exchange?

➤ Thanks for this kind and valuable suggestion.

Firstly, the experimental results show that 54% yield of desired product can be obtained without NaI additive, while the yield can be enhanced to 91% yield with two equivalents of NaI, which implies that the amount of NaI will affect the product yield. Secondly, our calculations indicate that the occurrence of halide anion exchange is thermodynamically driven with reaction energy of -3.4 kcal/mol when NaI is present in excess. Furthermore, due to the strong electron donating ability of I^- , the resulting electron-rich binuclear Ni center is conducive to enhance the nucleophilic ability of the alkenyl, which has also been further confirmed by NBO charge. As shown in the figure below, the transition state energy barrier of imine insertion mediated by iodide intermediates (**³ts2-S**) is 3.0 kcal/mol lower

than that mediated by chlorides (${}^3\text{ts}_{2\text{Cl-S}}$), which will enhance the activity of the reaction.

Comment 3: Other minor concerns relate to the energy differences between the four diastereomeric transition states, especially between ts_{2-S} and $\text{ts}_{2'-R}$, which are only 1.7 kcal/mol apart—a bit short to fully explain the high enantioselectivity. According to the authors' own calculations, the Boltzmann distribution yields a computed 87% ee (14:1), which is not as close energetically to the experimental 96% ee (49:1) as the authors suggest. This level of disagreement is understandable, given the nature of the species and methods used, but the agreement should not be oversold.

- We apologize for the confusion caused by our mistake. In fact, to simplify the model, we chose imine substituted with methylsulfonyl (Ms) group as reactants for mechanism studies (product **5** in Fig. 2). Calculated diastereomeric transition states energy difference between ${}^3\text{ts}_{2-S}$ and ${}^3\text{ts}_{2'-R}$ is 1.7 kcal/mol and the predicted ee value is 87% according to the Boltzmann distribution. For the reaction involving the imine substituted with methylsulfonyl (Ms) group, the experimental ee value is 76% (exp. $\Delta\Delta G = 1.2$ kcal/mol). Correspondingly, we made corrections for the relevant descriptions of “Considering the Boltzmann distribution, the predicted ee value is 87% (exp. 76%).” in the main text.

Comment 4: Finally, intermediate **Int-5** in Figure 7 should contain two iodine atoms attached to the Ni–Ni complex.

- These are two iodine atoms attached to the Ni–Ni complex. The other iodine atom is in the shaded area.

General Comments: These concerns do not affect the overall high quality of the work and manuscript. I therefore recommend publication of the article after minor revision, based on my previous comments.

- We appreciate this referee for the positive comments.

Reply to comments by Referee 3

General Comments: Although asymmetric reductive addition of imines and aldehydes has been in development for many years, this strategy is still limited to substrates with different modes of substitution. Synthetic chemists are pursuing and expecting a more general approach. Faced with this problem, Tao reported an efficient dinickel catalytic system with high substrate compatibility is achieved. We were surprised by the significant e.r. value advantage of this system compared to mononuclear nickel. The mechanism study and DFT study revealed that this selectivity comes from the fact that the binuclear nickel can bind the substrate tightly to the reaction core through additional Ni-substrate interaction, reducing its spatial freedom, and the asymmetric ligand-substrate interaction ultimately determines its selectivity. However, we think that there are still problems in DFT calculation, and we hope author revise it and then publish it on Nat. Commun.

- We appreciate this referee for the positive comments and valuable suggestions for the revision of the manuscript.

Here are our suggestions:

Comment 1: The PES of the exchange of NaCl and NaI is marked as -3.2 kcal/mol. However, due to the absence of the SPE and coordinates of I source and Cl source in SI, we cannot deduce this energy, and there is also a lack of reasonable literature citations to show that this energy decreases.

- We have supplemented the SPE and coordinates of NaCl and NaI in SI. Considering the quintet state mentioned by the reviewer in Comment 4, the

relatively stable quintet state $^5\text{int}2$ results in an energy of -1.2 kcal/mol for the ion-exchange step, which can be deduced through the thermodynamic data in Table S13. Finally, considering the energy correction caused by the excess NaI, the ion-exchange energy is -3.4 kcal/mol. We have made corresponding corrections for the reaction potential energy surfaces in SI (Figure S2)

Comment 2: Considering that this reaction can be carried out without sodium iodide, the need for iodide to replace chloride in this step is questionable, and it should be explained whether the free energy change in $\text{ts}2$ is higher in the case of chlorine-coordination.

➤ Thanks for this kind and valuable suggestion.

Firstly, the experimental results show that 54% yield of desired product can be obtained without NaI additive, while the yield can be enhanced to 91% yield with two equivalents of NaI, which implies that the amount of NaI will affect the product yield. Secondly, our calculations indicate that the occurrence of halide anion exchange is thermodynamically driven with reaction energy of -3.4 kcal/mol when NaI is present in excess. Furthermore, due to the strong electron donating ability of I^- , the resulting electron-rich binuclear Ni center is conducive to enhance

the nucleophilic ability of the alkenyl, which has also been further confirmed by NBO charge. As shown in the figure below, the transition state energy barrier of imine insertion mediated by iodide intermediates (${}^3\text{ts2-S}$) is 3.0 kcal/mol lower than that mediated by chlorides (${}^3\text{ts2}_{\text{Cl-S}}$), which will enhance the activity of the reaction.

Comment 3: Are there imine-coordinated intermediates before ts2 ? Its free energy may be lower than -30.9 kcal/mol, which is crucial to understanding the step energy of 1,2-addition with the dinuclear nickel species.

- Thanks for this kind and valuable suggestion. According to the Intrinsic Reaction Coordinate (IRC) results of the transition state ${}^3\text{ts2}$, we obtained that the energy of the reaction precursor for the transition state ${}^3\text{ts2}$ was -29.3 kcal/mol, whose energy is 1.6 kcal/mol higher than that of int3 . Therefore, this intermediate does not affect the energy barrier of the 1, 2-addition process.

Comment 4: The paper cited in SI (ACS Catal. 2017, 7, 7, 4796–4804) considers a quintet path under a similar system. Whether this path is possible in the reaction presented in this paper, we think it is necessary to add DFT calculations or related literature to illustrate this point.

- Thanks for this kind and valuable suggestion.
We have calculated the quintet state reaction paths and made corresponding corrections for the reaction potential energy surfaces in Figure S2. The results indicate that the quintet state is unfavorable during the oxidative addition process of C-Cl bond, although slight stability could be found in the generated bridging

species **int2**. In the imine migration insertion, the triplet state is still dominant in reaction paths, indicating enantiodetermining step will not be affected by other spin states, and thus our conclusions will not change.

Here are two minor problems:

Comment 5: **3int5** is missing in the potential energy surface.

- We apologize for misunderstandings caused by inappropriate intermediate labeling in Figure 7. In fact, our potential energy surface only calculates the key reaction steps, and it does not involve the final reduction process and int5. To avoid this misunderstanding, we have modified the **3int5** to **int5** in the Figure 7.

Comment 6: Many intermediates and transition states in Figure 6 lack ΔE value and have unreasonable relative energy heights (an example: “-0.3” is higher than “0”)

- Thanks for this kind and valuable suggestion. Due to the entropy effect, the energy of the transition state **3ts1** is slightly lower than that of the intermediate **int1**. To represent this point, we have marked the corresponding electronic energy. As for the entire potential energy surface, it is mainly based on ΔG , which contains the entropy effect and the solvation effect. To enhance readability, we made

modifications to Figure 6 and explained electronic energies in parentheses in the caption.